# Multi-metal electrohydrodynamic redox 3D printing at the submicron scale

Alain Reiser [1], Marcus Lindén [1], Patrik Rohner [2], Adrien Marchand [3], Henning Galinski [1], Alla S. Sologubenko [1], Jeffrey M. Wheeler [1], Renato Zenobi [3], Dimos Poulikakos [2] & Ralph Spolenak [1]

An extensive range of metals can be dissolved and re-deposited in liquid solvents using electrochemistry. We harness this concept for additive manufacturing, demonstrating the focused electrohydrodynamic ejection of metal ions dissolved from sacrificial anodes and their subsequent reduction to elemental metals on the substrate. This technique, termed electrohydrodynamic redox printing (EHD-RP), enables the direct, ink-free fabrication of polycrystalline multi-metal 3D structures without the need for post-print processing. On-the-fly switching and mixing of two metals printed from a single multichannel nozzle facilitates a chemical feature size of <400 nm with a spatial resolution of 250 nm at printing speeds of up to 10 voxels per second. As shown, the additive control of the chemical architecture of materials provided by EHD-RP unlocks the synthesis of 3D bi-metal structures with programmed local properties and opens new avenues for the direct fabrication of chemically architected materials and devices.

[1] Laboratory for Nanometallurgy, Department of Materials, ETH Zürich, CH-8093 Zürich, Switzerland. [2] Laboratory of Thermodynamics in Emerging Technologies, Department of Mechanical and Process Engineering, ETH Zürich, CH-8092 Zürich, Switzerland. [3] Laboratory of Organic Chemistry, Department of Chemistry and Applied Biosciences, ETH Zürich, CH-8093 Zürich, Switzerland. Correspondence and requests for materials should be addressed to R.S. (email: ralph.spolenak@mat.ethz.ch)

# 3D

printing revolutionises the way we design and fabricate. Today, additive manufacturing (AM), i.e. building objects in a voxel-by-voxel fashion, is an established extension of conventional manufacturing techniques. Applications range across all fields and length scales, from printed buildings[1] and bones[2] to microphysiological devices[3] and nanometric metamaterials[4]. In the last decade, major efforts have resulted in AM's superior geometrical abilities. Presently, two new frontiers of AM are the integration of functionality and locally optimised materials properties within a single printed object[5–8], and the downscaling of additive techniques to the micro- and nanoscale[4,9]. At this interface, printed lattice materials with properties tuned by their geometrical architecture have already opened a wide range of applications, including new classes of mechanical and optical metamaterials[4,10,11]. Now, emerging microscale AM methods with multi-material capabilities have started to extend the additive control to the local chemical composition, i.e. the chemical architecture, paving the way to additively manufactured batteries[12,13], thermocouples[14], bi-material metamaterials[15] and materials with locally tailored electrical resistance[16], chemical reactivity[17] or microstructures[18]. Yet, to see small-scale AM unfold its full potential for the fabrication of multi-material devices and chemically heterogeneous materials, major challenges need to be addressed: first, common multi-nozzle approaches enforce extensive practical limits to the complexity of the 3D chemical architecture[14,17,19]; second, as-deposited properties of inorganic materials, mostly dispensed as nanoparticle inks, are often far from those demanded in microfabrication[9,20], and the hence required post-print processing largely complicates many materials combinations.

Here, we introduce a microscale multi-metal AM technique called electrohydrodynamic redox printing (EHD-RP). This ink-free method overcomes the previously mentioned limitations of small-scale multi-material AM for the case of metals, as it enables the direct printing and mixing of multiple, high-quality metals from a single nozzle. EHD-RP combines the high spatial resolution of electrohydrodynamic (EHD) printing[21,22] with the in situ generation and deposition of metal ions from sacrificial anodes[23,24]. We demonstrate that the combination of submicron geometrical feature size and fast modulation of the printed chemistry offers unmatched control of the 3D chemical architecture of printed structures and enables tuning of local properties through local alloying at the submicron scale.

## Results

**Electrohydrodynamic redox printing (EHD-RP).** EHD-RP utilises basic electrochemistry to synthesise metallic deposits: in situ dissolution of a metal anode $M^0$ immersed in a liquid solvent (e.g. acetonitrile) generates solvated metal ions $M^{z+}$ inside the printing nozzle (Fig. 1a). These ions $M^{z+}$ are ejected towards the substrate, where they are reduced to form the metallic deposit $M^0$ (see Supplementary Discussion 1 and 2 for more details regarding the growth mode). The ion source is quasi-infinite, as the volume of the anode is many orders of magnitude larger than the printed volumes. The emission of ions is accomplished by EHD ejection of ion-loaded solvent droplets from the orifice of the printing nozzle[25]. An applied DC voltage of 80–150 V drives the EHD ejection of droplets and at the same time ensures a sufficiently high anodic surface potential for the dissolution of the source electrode. Direct printing onto metallic and semiconductive substrates as well as indirect printing across insulators is accessible with this concept (Supplementary Fig. 1).

The use of sacrificial metal electrodes combined with the localised electrochemical reduction of the generated ions on the substrate makes our work distinct from existing AM concepts in general[9], and more specifically, from previous demonstrations of ink-based EHD micro- and nanoprinting of metals[19,22,26]. Yet, sacrificial anodes as precursors for metal ions are well established for solution-based synthesis[27,28] and electrospraying of metal ions and particles[23,24,29,30]. EHD-RP differs from these experiments by enabling highly localised electrochemical growth of dense materials as well as the continuous modulation of the deposited chemistry, as opposed to the deposition of isolated particles[23,24].

**Multi-metal and alloy printing from a single nozzle.** A key feature of EHD-RP is the simultaneous printing of multiple metals from a single multichannel nozzle (Fig. 1a). To grow two different metals A and B, a wire of A and a wire of B are each placed in one of the compartments of a two-channel nozzle (Fig. 1b). If the positive voltage is applied to one of the wire electrodes only, then only one ion species $A^{z+}$ or $B^{z+}$ is created and ejected, and A or B is printed. If both electrodes are biased simultaneously, an alloy A–B is deposited. Figure 1d, e demonstrates this principle for a Cu and a Ag electrode: mass spectra of the ejected ions reflect the elemental nature of the biased electrodes (Fig. 1d), and energy-dispersive X-ray (EDX) spectroscopy confirms the corresponding chemical composition of the printed structures (Fig. 1e).

**On-the-fly variation of the printed chemistry.** The prompt, on-the-fly control of ion generation facilitates fast variation of the printed chemistry. Figure 2a shows mass spectrometry (MS) transient ion currents recorded upon switching the voltage between a Cu and a Ag electrode at different frequencies. Alternating ion-current pulses as narrow as 100 ms are clearly resolved. Selectivity of the ON ion species over the OFF species is high, although a spike of OFF ions ejected at the beginning of each pulse indicates purging of leftover ions (Supplementary Fig. 5). Upon printing, the individual pulses of $Cu^+$ and $Ag^+$ ions translate into chemical modulations, and are resolved up to the minimal pulse width of 100 ms (Fig. 2b). However, mixing of the individual chemical signals is evident for intervals smaller than 1 s. Since transient MS currents indicate more selective switching, mixing of the signals is at least partially attributed to the limited lateral resolution of the EDX line scans. In practice, ion pulses of 100 ms were used for printing, and chemical features smaller than 400 nm were printed.

The rapid switching of the printed metals enables the direct synthesis of chemically heterogeneous 3D geometries with high spatio-chemical resolution. As one of two examples, pillars with individual numbers of Cu and Ag segments were grown by alternating the anodic electrode potentials at intervals of 0.5–1 s while printing in a serial, point-by-point manner (Fig. 2c and Supplementary Fig. 4). As a second example, a Cu-wall with an embedded chemical image of the letters 'Ag' printed in silver demonstrates the realisation of an intricate chemical architecture by switching between two metals upon printing in a layer-by-layer mode (Fig. 2d, Supplementary Movie 1). Using a single nozzle, the complexity of the chemical sequence is increased without a penalty in printing speed and added alignment errors typical for multi-nozzle systems. Nonetheless, although the lateral misalignment upon switching is reduced to a minimum, a slight shift between the two metals is often caused by the asymmetry of the two-channel nozzle (Fig. 2b, c).

**Spatial resolution, microstructure and materials properties.** Geometrical features printed by EHD-RP are well within the submicron range (Fig. 3). The spatial resolution is 250 nm for layer-by-layer printing modes (Fig. 3b), and in-plane features smaller than 100 nm have been printed (Fig. 3c). Out-of-plane wires with aspect

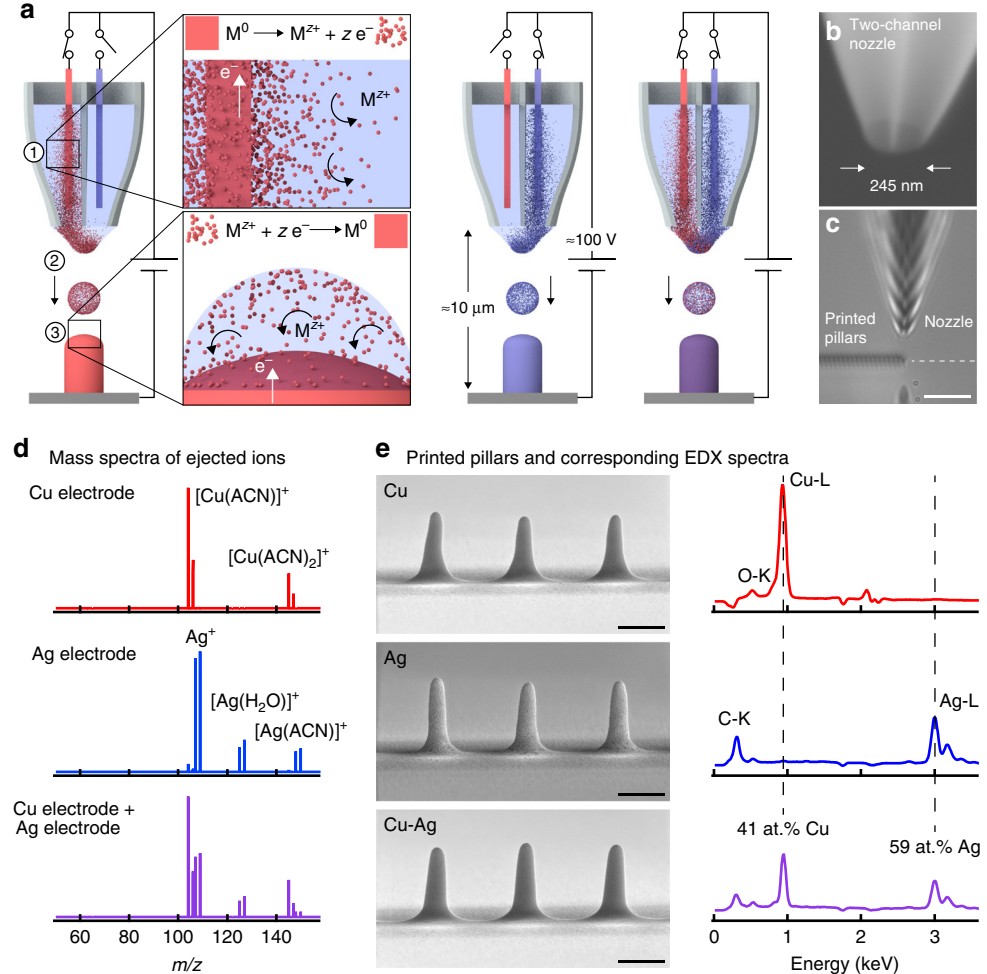

**Fig. 1** Electrohydrodynamic redox printing (EHD-RP). **a** Working principle: (1) Solvated metal ions $M^{z+}$ are generated within the printing nozzle via electrocorrosion of a metal electrode $M^0$ immersed in a liquid solvent. (2) Ion-loaded solvent droplets are ejected by electrohydrodynamic forces. (3) Upon landing, $M^{z+}$ ions are reduced to zero valence metal $M^0$ through electron transfer from the substrate. Switching the oxidative voltage between different electrodes in a multichannel nozzle enables on-the-fly modulation of the printed chemistry (Schematics not drawn to scale: typical dimensions of the electrode wire are 100 µm × 2 cm). **b** Typical two-channel nozzle. **c** Optical micrograph of the printing process. Scale bar: 10 µm. **d**, **e** Printing Cu, Ag and Cu–Ag from a single, two-channel nozzle. **d** Mass spectra of ejected ions when biasing the Cu electrode, the Ag electrode, or both electrodes immersed in acetonitrile (ACN). **e** Printed Cu, Ag and Cu–Ag pillars with corresponding energy-dispersive X-ray (EDX) spectra reflecting the chemical nature of the respective source electrode (background subtracted). The C–K and O–K peaks likely originate from residual solvent and minor oxidation, respectively. The Cu and Ag contents of the Cu–Ag pillars are given in at.% normalised to the total Cu + Ag signal. Scale bars: 500 nm. For imaging conditions of micrographs, see Supplementary Table 1

ratios of $10^2$–$10^3$ (Fig. 3d and Supplementary Fig. 6) and overhangs of up to 90° (Fig. 3e) are accessible with point-by-point strategies, while more complex geometries can be printed with a layer-by-layer approach (Fig. 3f). Typical in- and out-of-plane printing speeds are on the order of 1 µm s$^{-1}$ for minimal feature sizes of 150–250 nm, i.e. approximately 5 voxels s$^{-1}$ when normalised by the X–Y feature size. Higher growth rates are physically possible, but the maximum practical out-of-plane printing speed is currently on the order of 10 voxels s$^{-1}$. A further increase in growth speed is presently hampered by a sessile solvent droplet on the substrate that results at higher ejection voltages (Supplementary Fig. 8). In general, the printing speed is an order of magnitude higher than for other electrochemical microscale AM techniques[31,32] and enables printing of structures several hundred micrometers in width (Supplementary Fig. 7 and Supplementary Movie 2). However, geometrical complexity and fidelity are currently not as high as those offered by some alternative single-metal techniques[9] —a shortcoming that is shared amongst all EHD-based

microprinting techniques, but could be ameliorated by the use of a growth feedback[32–34] and, as demonstrated below, the use of support structures.

The electrochemical growth process offers functional as-deposited materials properties without the need for additional processing. The microstructure of the printed metals is nanocrystalline and dense (>90%, Fig. 3g and Supplementary Fig. 9) and free of oxides when printed in Ar (Supplementary Fig. 10). A strength of 1–1.5 GPa for Cu pillars competes with the highest literature values for nanocrystalline Cu (Supplementary Fig. 12), and lines printed onto electrical insulators achieve an as-deposited electrical conductivity of up to 0.12× bulk conductivity of Cu (line width: 200 nm, Supplementary Fig. 13). In general, the grain size, texture and defect-density of the deposited material are a function of the applied electric field during printing (Supplementary Fig. 11), offering the potential to adjust printed microstructures to the properties required by a given application (e.g. high conductivity versus high strength).

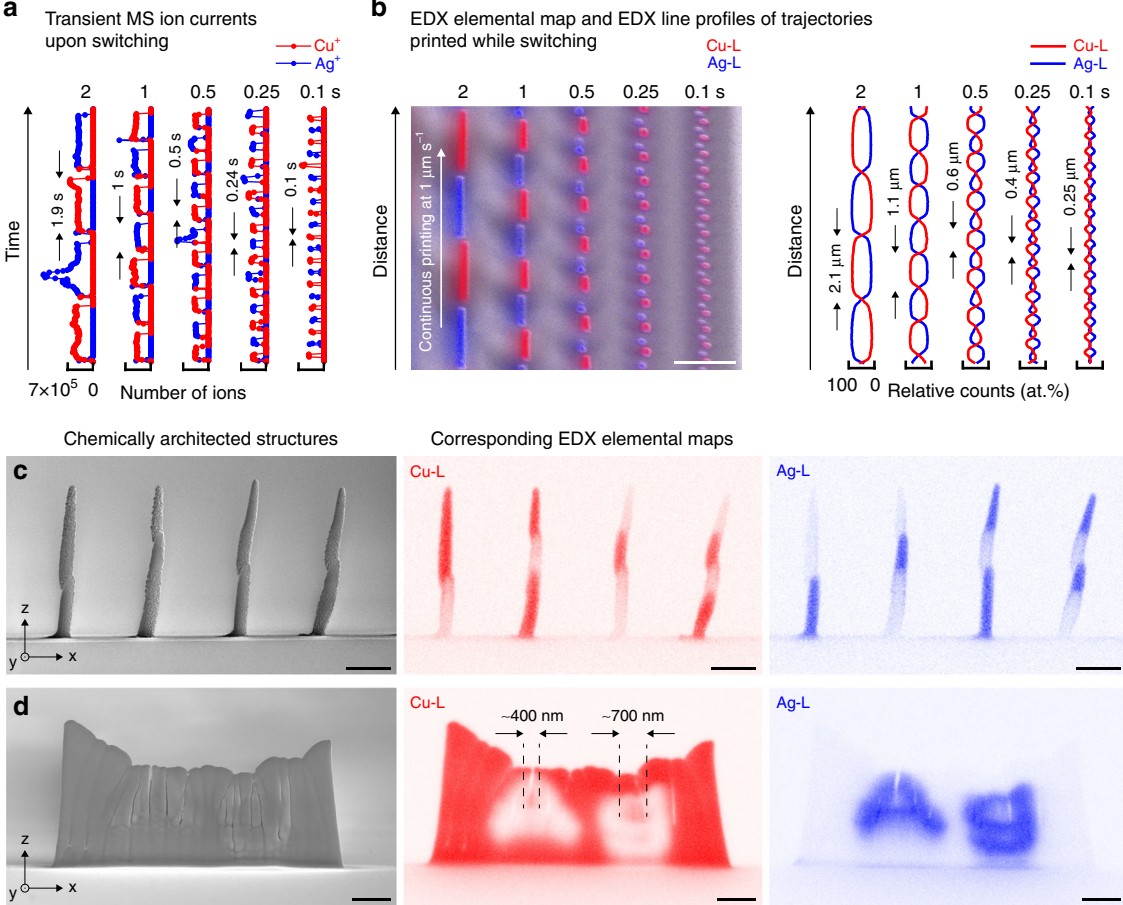

**Fig. 2** Additive control of the chemical architecture with a single nozzle. **a, b** Fast switching between two metals printed from a two-channel nozzle. **a** Summed mass spectrometry (MS) ion currents of $Cu^+$ (red) and $Ag^+$ (blue) cations ejected upon switching the anodic voltage between a Cu and a Ag electrode at different intervals. Switching between two ejected ion species is highly selective. **b** Overlaid SE micrograph and EDX elemental map of trajectories printed with the same switching profile as in (**a**) (Cu-L signal, red, and Ag-L signal, blue). The corresponding EDX line profiles show that the switching between Cu and Ag is resolved up to the smallest pulse width. Scale bar: 2 μm. **c, d** Examples of chemically heterogeneous structures printed using a single nozzle. **c** Sequence of pillars with different numbers of Cu and Ag modulation periods. Scale bars: 1 μm. **d** Out-of-plane Cu wall with the letters 'Ag' embedded in silver, printed with a continuous layer-by-layer printing mode (Supplementary Movie 1). Scale bars: 1 μm. For printing parameters, see Supplementary Table 1

**Tailored local properties via tailored local chemistry**. Finally, the additive control of the chemical architecture provides the means to extend the classic approach of tuning properties by alloying to the individual portions of a printed structure, enabling the deterministic programming of local materials properties through the architecting of the local composition. Indeed, we demonstrate two examples of chemically heterogeneous structures that express local variations in properties based on tuning of the local electrochemical nobility. First, we employ selective etching of printed Cu support structures to extend the range of accessible geometries to bridging structures (Fig. 4a–c). Second, we utilise dealloying of co-printed Cu–Ag segments (45–50 at.% Cu) to fabricate pillars with local modulations in nanoscale porosity and thus a step-function in mechanical strength (Fig. 4d–h). The latter is showcased by the significantly increased plasticity within the softer, porous segments upon bending of the pillars (Fig. 4f, Supplementary Movie 3). In both examples, the combination of two metals of different electrochemical nobility within one printed structure enables applications that are inaccessible with the individual materials. In general, a variety of mechanical, optical and reactive properties can be accessed with the three currently printable metals Cu, Ag and Au (for Au, see Supplementary Fig. 2). Additionally, the large range of sacrificial metal anodes employed in solution-[27,28] and spray-based synthesis[30] suggests that the range of printable materials can be extended towards other metals and non-metals, broadening the tuneable property space.

In summary, EHD-RP markedly advances the state of the art of multi-metal AM at small scales, offering unmatched control of the 3D chemistry of additively synthesised metals. As shown, the direct, ink-free process results in excellent mechanical and good electrical properties printed at competitive speeds and ambient pressures — readily permitting applications such as small-scale wire bonding, optical or mechanical metamaterials and printed sensors or actuators. Although we focused on three distinct metals, the approach can be extended to more complex materials combinations by using nozzles with more than two channels and by exploring synthesis protocols that grant access to additional metals and potentially non-metals, i.e. semiconductors, ceramics and polymers. Thus, EHD-RP holds the potential for unlocking unique routes for the bottom-up fabrication of chemically designed 3D devices and materials with locally tuned properties and a rational use of alloying elements. Such materials could find application in catalysis, active chemical devices, small-scale robotics and architected materials that go beyond single-material cellular designs.

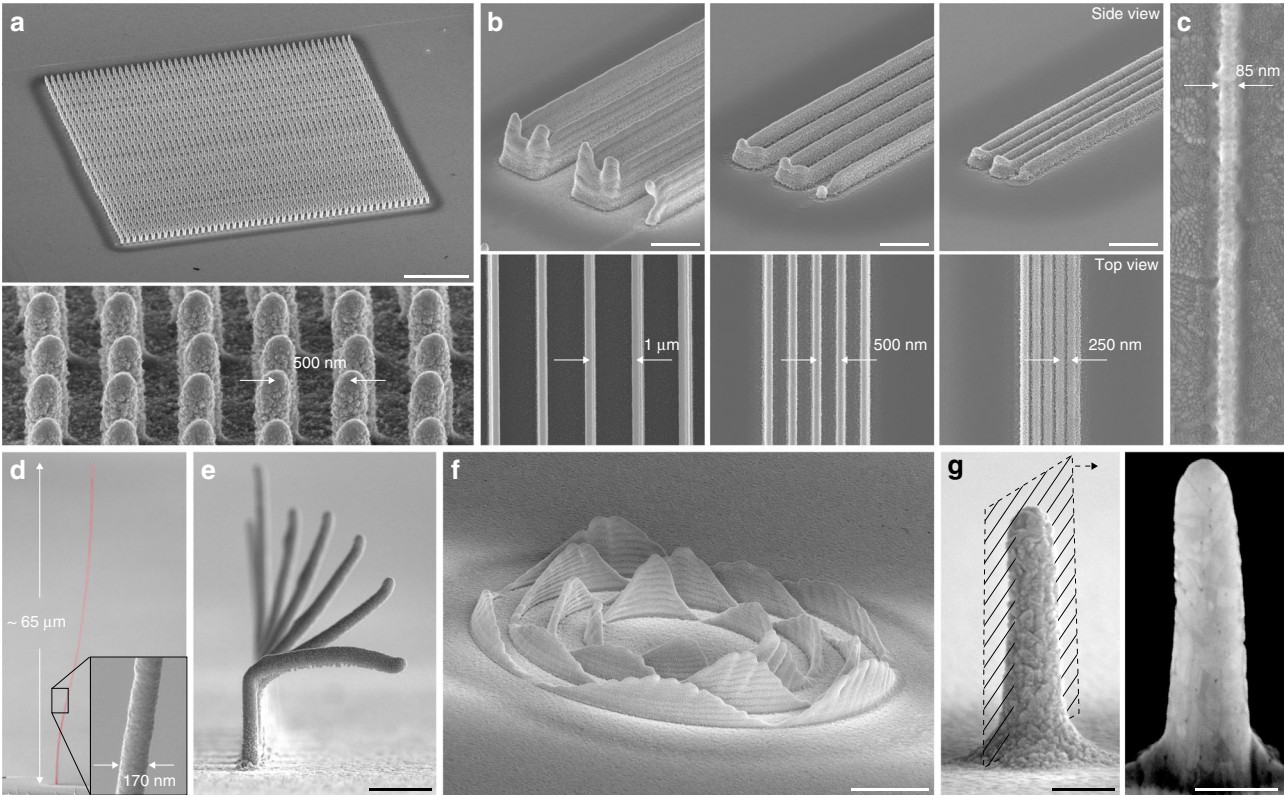

**Fig. 3** Geometrical performance and as-printed microstructure. **a** Array of 50 × 50 Cu pillars printed with a point-to-point spacing of 500 nm. Scale bar: 5 μm. **b** Walls printed at decreasing wall-to-wall spacing, with a minimum spacing of 250 nm. Height: ten layers for the leftmost image, three layers for the others. Scale bars: 1 μm. **c** Printed Cu line less than 100 nm in width. **d** Cu wire with an aspect ratio of approximately 400. **e** Overhangs formed by a lateral translation of the stage balancing the out-of-plane growth rate. The sequence of pillars was printed by increasing the respective in-plane translation speed towards the front pillar, with a maximum speed of 2.1 μm s$^{-1}$. Scale bar: 1 μm. **f** Concentric, out-of-plane sine waves printed with a layer-by-layer strategy. Scale bar: 2 μm. **g** As-printed Cu pillar and corresponding cross-section showing the dense, polycrystalline microstructure. Scale bars: 200 nm

## Methods

**Materials**. Single and two-channel printing nozzles were fabricated from quartz capillaries (Sutter Instrument) using the P-2000 micropipette puller system (Sutter Instrument). Pulling parameters for typical nozzle diameters are listed in Table 1. Nozzle diameters in the range of 160–200 nm facilitated the most focused ejection and were thus primarily used (Supplementary Fig. 3). Prior to printing, the nozzles were rinsed with acetonitrile (Optima, Fisher Chemical) and subsequently filled with acetonitrile using a glass syringe (Gastight #1010, Hamilton) equipped with a syringe-filter (20 nm pore-size, Anotop 10, GE Whatman) and a non-metallic needle (MF34G-5, World Precision Instruments). In general, the overall cleanliness of the system was of high importance. Especially organic contaminants can be problematic, since many of them may dissolve in acetonitrile and are consequently co-deposited with the metallic species. Carbon-based residues are more often observed when printing Au and Ag, while they are hardly an issue with Cu (compare EDX spectra in Fig. 1 and Supplementary Fig. 2). Further contaminants can be leached from the glass: the use of borosilicate nozzles resulted in unusually large quantities of sodium in the case of Ag. Switching to quartz glass significantly increased the purity and strength of the printed metals.

Substrates were usually either ITO coated glass slides (TIXZ 001, Techinstro) or Au thin-films deposited in our laboratory sputter facility by DC magnetron sputtering (PVD Products Inc.). 100 nm thick Au films (3 mTorr Ar, 200 W) on top of a 12 nm thick Ti adhesion layer (3 mTorr Ar, 250 W) were sputtered on (100)-Si wafers (SiMat). In general, the base pressure of the sputtering system was 10$^{-7}$ Torr, and a substrate rotation of 30 rpm was used to obtain uniform film thicknesses. Au substrates covered with an electrically insulating Al$_{2+x}$O$_{3-x}$ layer were prepared by depositing a 1 μm thick film of the oxide by reactive sputtering (100 sscm Ar, 5.75 sscm O$_2$, 5 mTorr) on previously prepared Au-substrates. Prior to reactive sputtering, the Al target was cleaned for 10 min at 300 W. For direct printing on a semiconductor substrate, an undoped Ge single crystal was used as-received ((123), 1GE 005E, Crystal GmbH, Berlin).

Metal wires were used as electrodes for printing. Before printing, Cu (0.1 mm dia., 99.9985%, Alfa Aesar) and Ag wires (0.1 mm dia., 99.997%, Alfa Aesar) were etched in pure nitric acid (Sigma Aldrich) for 20 s and 2 min respectively, and subsequently rinsed with deionised water and dried in air at ambient temperature.

Au wires (0.1 mm dia., 99.998%, Alfa Aesar) were either used as-received or were sonicated in acetonitrile before printing.

**Printing setup**. The main components of the printing setup were (Supplementary Fig. 16): a power source (B2962A, Keysight) for biasing the electrodes; mechanical relays (HE751, Littlefuse) for switching the high voltage between different electrodes; a printing nozzle; an electrically grounded substrate mounted on a three-axis piezo translation stage (QNP60XY-500-C-MP-TAS, QNP60Z-500-C-TAS, Aerotech); an optical microscope to observe the printing process. The piezo axes were mounted on an additional long-range axis (M112-1VG, PI) to enable stage translations larger than 500 μm. The nozzle was fixed to a manual three-axis micromanipulator-stage (HS 6, Märzhäuser Wetzlar GmbH) for coarse positioning. The mechanical relays were computer-controlled and interfaced with an Arduino UNO microcontroller board. The microscope was built from a ×50 objective-lens (LMPLFLN, Olympus), a CMOS camera (DCC1545M, Thorlabs), and a blue LED light source (LEDMT1E, Thorlabs). The optical axis of the lens was inclined 60° to the substrate normal. To print at low oxygen levels (200–1000 ppm), an acrylic chamber enclosing the printer was flushed with argon gas (4.8, PanGas). Oxygen- and humidity-levels were monitored with a gas sensor (Module ISM-3, PBI Dansensor) and a humidity sensor (SHT31, Sensirion). Usual relative humidity and temperature during printing were 30–50% and 20–25 °C, respectively.

**Printing parameters**. Typical electrical fields for printing were on the order of 10$^7$ V m$^{-1}$, with electrode voltages of 80–150 V and a typical nozzle-substrate distance of 5–10 μm. Throughout the text, we define the electric field as the voltage applied to the source divided by the tip–substrate distance. We use this oversimplified definition because a more precise description of the actual electric field was not attainable: the concentration of solvated ions and thus the solvent conductivity upon printing are unknown. Printing at low fields of 10$^7$ V m$^{-1}$ guaranteed highest resolution, low surface roughness and facile control of the printing process. In contrast, higher fields offered increased growth rates (Supplementary Fig. 7) and access to additional growth morphologies and microstructures (Supplementary Fig. 11). Yet, the high speed complicated control of the overall printing process. For

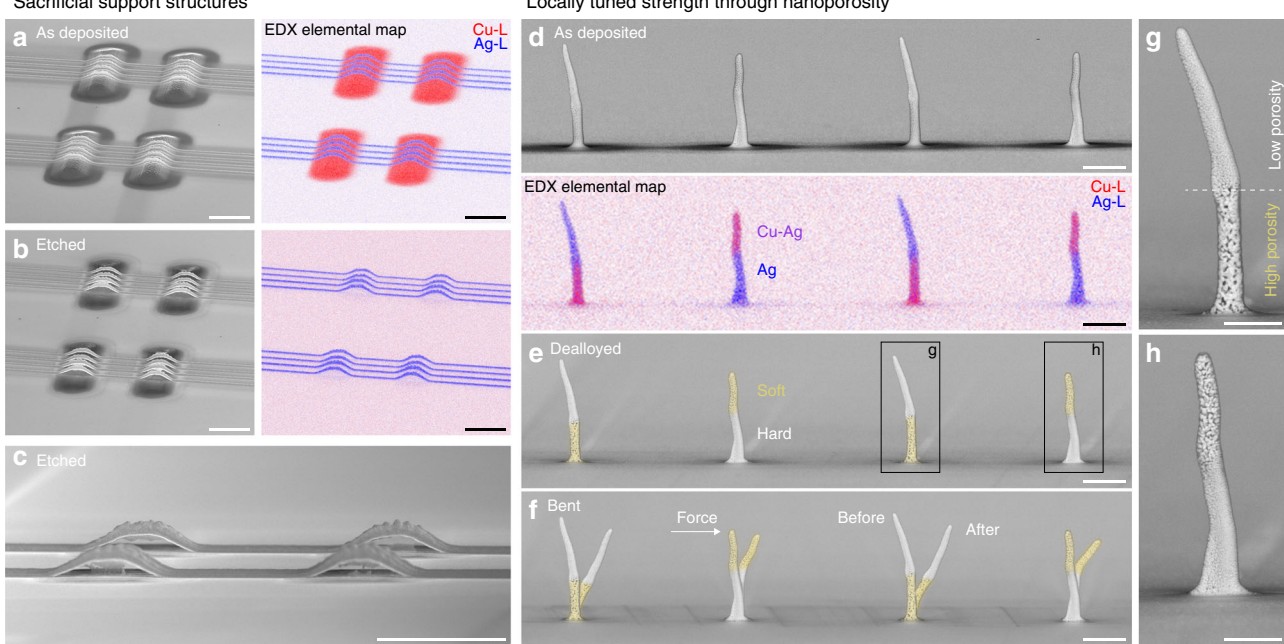

**Fig. 4** Tailored local materials properties via the design of the chemical architecture. Exploiting the dissimilar nobility of Cu and Ag for (**a–c**) printing of sacrificial support structures and (**d–h**) tuning of local porosity and thus local mechanical strength. **a** Ag lines printed on top of Cu support structures before and (**b, c**) after selective wet etching of Cu. Scale bars (**a–c**): 5 µm. **d** Bi-metal pillars printed with alternating segments of Ag and Cu–Ag before and (**e**) after selective dissolution of Cu. The dealloying of Cu from the Cu–Ag segments renders partially nanoporous Ag pillars with soft (porous, coloured in yellow) and hard segments. **f** The step-function in porosity modulates the local mechanical strength: upon bending of the pillars, the plastic deformation is pronouncedly confined in the porous segments (Supplementary Movie 3). Scale bars (**d–f**): 1 µm. **g, h** Close-up of the partially nanoporous pillars after dealloying. Scale bars: 500 nm

### Table 1 Nozzle pulling parameters

| Diameter [nm] | Glass | Line | Heat | Fil | Vel | Del | Pull | Loops |
|---|---|---|---|---|---|---|---|---|
| 1300 ± 100 | QF100- | 1 | 750 | 5 | 25 | 128 | 50 | |
| | 70-10 | 2 | 500 | 4 | 50 | 130 | 100 | 1 |
| 170 ± 20 | QF100- | 1 | 850 | 5 | 25 | 128 | 50 | |
| | 70-10 | 2 | 700 | 4 | 50 | 130 | 100 | 1 |
| 100 ± 20 | QF100- | 1 | 850 | 5 | 25 | 128 | 50 | |
| | 70-10 | 2 | 750 | 4 | 50 | 130 | 100 | 1 |
| 200 ± 30 | QT120- | 1 | 800 | 4 | 40 | 130 | 30 | |
| | 90-7.5 | 2 | 825 | 3 | 30 | 130 | 100 | 1 |

Pulling parameters for single (QF100) and two-channel (QT120) quartz nozzles. Heat, filament (fil), velocity (vel), delay (del) and pull refer to the five programmable parameters of the P-2000 micropipette puller system

a more detailed discussion of the printing speed, see Supplementary Note 6. Printing speeds and number of layers used to print the structures presented in the main text are listed in Supplementary Table 1.

When printing with multiple electrodes, the voltage needed to be adjusted for each electrode in order to match the individual growth rates. In general, it should be noted that all electrodes not in use had to be electrically floating, i.e. neither connected to electrical ground nor any of the high voltage outputs. In case multiple electrodes were printed at the same time, all electrodes had to be held at the same voltage. If these guidelines were not followed, parasitic currents between electrodes caused cross-contamination of the electrodes, because individual channels of the nozzle are connected across a liquid bridge at the nozzle tip. See Supplementary Note 4 for a more detailed discussion of individual growth rates and alignment of the two materials printed from a two-channel nozzle.

**Selective etching of Cu.** Selective etching of Cu from printed Cu–Ag structures was performed under potential control at 0.25 V versus a saturated calomel electrode in 10 vol.% aqueous phosphoric acid (Puriss p.a., >85%, Sigma Aldrich),

using an Autolab PGSTAT12 potentiostat (Metrohm) and a Pt counter-electrode. The etching solution was not agitated. Typical etching times were 45 s for sacrificial Cu support structures and 15 s to generate nanoporous Ag (Fig. 4). Samples were blow dried with air after etching.

**Analysis.** All MS experiments were performed on a Synapt G2S (Waters) instrument in positive ion mode using the time-of-flight mode (ion mobility was not activated). The extraction voltage was set to 150–400 V using an external power supply (B2962A, Keysight). The nozzle was positioned at a distance of 2–4 mm from the grounded MS cone. Spectra were recorded with an integration time of 1 s and summed over several acquisitions. For transient measurements, spectra were acquired at intervals of 20 ms. All transient MS data in Fig. 2a and Supplementary Fig. 5 correspond to the sum of the ion currents recorded for all species containing $Cu^+$ and $Ag^+$, and is presented as-recorded. The following peaks were summed: Cu: $m/z$ of 104, 106, 145 and 147, Ag: $m/z$ of 107, 109, 125, 127, 148 and 150.

Optical microscopy was conducted using a VHX-6000 optical microscope (Keyence) equipped with a VH-Z500R and a VH-Z20T lens. Analysis of geometry and chemical composition of the printed structures, as well as chemical mapping, were performed by scanning electron microscopy, using a Magellan 400 SEM (Thermo Fisher Scientific, former FEI) equipped with an Octane Super EDX-system (EDAX, software: Genesis, EDAX). Supplementary Table 1 reports tilt angles and detection modes of the microscopy images presented in the main text. EDX data were always recorded with 10 kV acceleration voltage. In general, atomic percentages of Cu and Ag were quantified by EDX and are stated relative to the total Cu–Ag signal to avoid any contributions of signals from the substrate. The EDX spectra in Fig. 1e are presented with background subtraction: all spectra of the pillars and of the substrate were normalised by the Au peak originating from the substrate, and the background spectra were subtracted from the spectra of the pillars. The elemental content profiles of Cu-L and Ag-L signals in Fig. 2b are normalised by the summed signal (at.%) of both lines. For the elemental content map in Fig. 4b (etched state), the intensity of the Cu-L signal (representing nothing but background noise of the detector) was scaled to match the background signal of the corresponding Ag-L map. For preparation of cross sections and TEM lamellae of printed pillars, a dual beam FIB (NVision 40, Zeiss) was used, employing typical $Ga^+$-ion-milling currents of 1 pA at 30 kV for final polishing. Transmission electron microscopy (TEM) analyses were performed on a Talos F200X TEM (Thermo Fisher Scientific, former FEI) operated at 200 kV in both, TEM and STEM imaging modes. STEM signals were recorded using bright field (BF STEM),

low-angle annular dark field (LAADF) and high angle annular dark field (HAADF) detectors simultaneously with a probe size of 0.7 nm.

Electrical resistance measurements were performed in a dual beam FIB microscope (NVision 40, Zeiss) using a Keithley 6430 source-meter and a Kleindiek W micromanipulator needle to establish an electrical contact to Pt pads that were deposited onto the printed line (for more detail, see Supplementary Fig. 13). Samples for electrical measurements were printed from grounded, 75 nm thick Ag electrodes (thermally evaporated, base pressure 3.2 mbar, 1 Å s$^{-1}$, Kurt J. Lesker Company) onto p-type (100) Si wafers (SiMat) covered with a 50 nm thick thermal oxide and a 50 nm thick Si$_3$N$_4$ diffusion barrier.

Microcompression testing was performed using an ultra-nanohardness tester (UNHT, Anton Paar Tritec SA, formerly CSM Instruments) and an in situ SEM indenter (Alemnis GmbH)[35] installed in a Vega3 SEM (Tescan). Both indenters were fitted with a 8 μm diameter diamond flat punch (Synton-MDP). Pillars were compressed either under displacement control at a rate proportional to the pillar's height to produce a strain rate of 0.01 s$^{-1}$, or under load-control at a rate of 0.5 mN min$^{-1}$, corresponding to a strain rate of 0.01–0.05 s$^{-1}$. To achieve a clear plastic yield point that is not influenced by the rounded top of the pillars, the top portions of the pillars were flattened by FIB-polishing prior to testing, using an ion-milling current of 10 pA. Bending of the pillars in Fig. 4 was performed by horizontally applying a displacement to the uppermost ≈300 nm of the pillars using a sharpened tungsten needle (Picoprobe, GGB Industries Inc.). The needle was translated by the Alemnis indenter system, and the experiments were observed in situ in a LEO 1530 SEM (Zeiss) (see Supplementary Movie 3 for details). The applied lateral forces were not measured.

Out-of-plane growth rates were measured and averaged for 50 pillars per reported speed value. The speed was derived by measuring height $h$, diameter $d$ and printing time $t$ for each individual pillar. The approximate number of voxels per pillar was defined as $n_{voxel} = h/d$, and the resulting voxel-speed, i.e. a growth rate normalised by the feature size, as $v_{voxel} = n_{voxel}/t$.

## Data availability

All data needed to evaluate the conclusions in the paper are present in the paper and/or the Supplementary Materials. Additional data related to this paper may be requested from the corresponding author.

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

## Acknowledgements

The authors thank S. Danzi, R. Frey, S. Ganzeboom, V. Vojtěch (all Laboratory for Nanometallurgy, ETH Zürich) and C. Murer (Laboratory for Magnetism and Interface Physics, ETH Zürich) for experimental support, and T. Kyburz (Laboratory for Nano-metallurgy, ETH Zürich) for machining setup components. The authors gratefully thank D. Momotenko (Laboratory of Biosensors and Bioelectronics, ETH Zürich) and T. Suter (Laboratory for Joining Technologies and Corrosion, EMPA Dübendorf) for fruitful discussions on the topic of nozzle fabrication and access to their nozzle-pullers. This work was funded by Grant no. ETH 47 14-2, and partially supported by Grant nos. SNF 200021-146180 and SNF 200020-178765. Electron-microscopy analysis was performed at ScopeM, the microscopy platform of ETH Zürich.

## Author contributions

A.R. and R.S. devised the concept. R.S. supervised the project. D.P. supervised P.R.'s contribution to the work. M.L., A.R. and H.G. built the printing setup, with input of know-how on the setup and basics of EHD and NanoDrip printing from the laboratory of D.P. (P.R.). M.L. wrote the printing software. M.L. and A.R. designed and performed printing experiments reported here, preliminary feasibility experiments were performed by P.R. and M.L. in the laboratory of D.P. A.M., A.R. and M.L. designed and performed MS experiments in the laboratory of R.Z. A.M. performed MS data analysis. A.R. and M.L. provided SEM and FIB analysis. A.S.S. performed TEM analysis. J.M.W. and A.R. performed micromechanical analysis. A.R., M.L., H.G. and R.S. validated the results on a regular basis, while all authors discussed the results. The work builds on results of M.L.'s master thesis. A.R. wrote the original paper draft and visualised the data. All authors reviewed and commented on the draft.

## Additional information

**Competing interests:** D.P. is the cofounder of SCRONA. The remaining authors declare no competing interests.

