## [Peer Review File · Nature Communications]

REVIEWERS' COMMENTS:

Reviewer #1 (Remarks to the Author):

This paper describes a new technique called “electrohydrodynamic redox” 3D printing, which is capable of printing metals at sub-micron length scales. The process works by anodizing a metal source (e.g. a wire) inside a capillary nozzle, which is then electrohydrodynamically sprayed onto a substrate due to the potential difference between the nozzle and conductive substrate. Upon reaching the substrate, the metal ions reduce to form metal. By using two metal sources, it is possible to deposit (or co-deposit) different metals. The work shows several demonstrations including co-printing of two metals (Ag and Cu), selective removal of one metal, high aspect ratio structures, and porous structures. The work is very beautiful and well-done. Since it is a methods paper - and the method works - I do not have any technical concerns. I think it is a good fit for Nature Communications since it represents a significant advance over other examples of reduction-based printing.

Reviewer #2 (Remarks to the Author):

The current manuscript is a revised version of the one I reviewed three months ago. The authors in their response and changes made to their revised manuscript have addressed all the questions and comments raised by this reviewer thoroughly and satisfactorily. It is the opinion of this reviewer that the revised manuscript merits publication in Nature Communications.